# Early-Onset Candidemia in Adult Intensive Care Units

**DOI:** 10.3390/diagnostics15111402

**Published:** 2025-05-31

**Authors:** Christina Mouratidou, Kalliopi Tsakiri, Vasiliki Dourliou, Alexandra Marneri, Maria Stougianni, Efstathios Pavlidis

**Affiliations:** 1Department of Adult Intensive Care Unit, Ippokrateio General Hospital, 54642 Thessaloniki, Greece; kalliopi.tsakiri@gmail.com (K.T.); vicky_dourliou@hotmail.com (V.D.); alexia.mrn@gmail.com (A.M.); mstougianni@gmail.com (M.S.); 2Second Propedeutical Department of Surgery, Medical School, Aristotle University of Thessaloniki, Ippokrateio General Hospital, 54642 Thessaloniki, Greece; pavlidis.md@gmail.com

**Keywords:** early-onset candidemia, *Candida* species, intensive care unit, critically ill, adult patients, antifungal agents, invasive candidiasis, rapid diagnostics, antifungal stewardship

## Abstract

Candidemia is a significant cause of morbidity and mortality among critically ill patients. Early-onset candidemia is characterized by occurring within the first seven days after admission to the Intensive Care Unit and presents several important challenges regarding its management. Risk factors may vary among patients with early- and late-onset infection, while clinical manifestations are generally non-specific and covered by the underlying disease and co-morbidities. Diagnosis and appropriate therapy are frequently delayed, with a high risk of progression to invasive, deep-seated infections, leading to rapid clinical deterioration. Management strategies to optimize the approach for patients with early-onset candidemia include the use of both conventional and novel diagnostic techniques, the initiation of appropriate antifungal therapy, administration of an adequate dose, daily evaluation of clinical response, de-escalation treatment whenever possible, and early discontinuation. Incorporating an antifungal stewardship program in clinical practice is essential in order to achieve the best clinical outcomes. Based on a review and analysis of the available literature, this article provides a thorough update on the risk factors, clinical characteristics, diagnostic methods, and management of early-onset candidemia in adult critically ill patients.

## 1. Introduction

Fungi represent the fourth most common cause of infections among patients in intensive care units (ICUs) worldwide, resulting in increased length of hospital stay, raised healthcare costs, and worse outcomes [1]. The major fungi implicated in the critical care environment are *Candida* and *Aspergillus* species, followed by less common species such as *Histoplasma*, *Blastomyces dermatitidis*, *Cryptococcus neoformans*, *Zygomycetes* and *Pneumocystis jirovecii* in certain groups of patients [2].

Alongside, bloodstream infections (BSIs) are a major cause of infectious disease morbidity and mortality among ICU patients. *Candida* species are responsible for more than 90% of fungal BSIs. Isolation of *Candida* species from blood cultures from a peripheral or central line is defined as candidemia. While *Candida albicans* (*C. albicans*) was the most clinically significant aetiological factor of fungal BSIs for decades, the increased rate of infections caused by other *Candida* species has altered the epidemiological pattern of candidemias in recent years [3,4,5]. Candidemia is most frequently observed in hospitalized patients, with a median appearance time of 3–4 weeks after hospital admission [6]. Although, there are several reports of candidemia developed less than 48 h after hospitalization. Early-onset candidemia is commonly characterized by occurring within the first 7 days of ICU admission, in contrast with late-onset infections that develop later during the hospital stay [7]. Early-onset candidemia has different clinical characteristics and risk factors compared to late-onset, with a high risk of mistreatment and the development of invasive, deep-seated infections [5,8].

In this review, we discuss characteristics of early-onset candidemia in adult intensive care units, demonstrating common pathogens, risk factors, clinical assessment, available diagnostic techniques, treatment options, and preventive strategies.

## 2. Common Pathogens

*Candida* is a large genus of budding, white, colony-forming yeasts, classified in the subphylum Saccharomycotina. Worldwide, *Candida* constitutes the largest group of medically important yeasts and is the predominant cause of fungal infections [9]. They are commensal microorganisms that are normally presented in human flora, specifically in the mucosa of the oropharyngeal cavity, gastrointestinal and vaginal tract, and skin [10]. Different mechanisms have been developed by *Candida* pathogens in order to survive within the human host, such as morphogenic transitions, metabolic flexibility, environmental pH, oxygen levels, temperature adaptations, and the ability to overcome micronutrient restrictions. Fifteen *Candida* species have been described as causative factors of human fungal infections, of a total of approximately 200 species [11,12].

In adults, early-onset candidemia is often linked to pre-existing factors that disturb the mucosal barrier and jeopardize immune response. The most common pathogens responsible for early-onset BSIs in critical care patients are:

*Candida albicans. C. albicans* is one of the most frequent causes of candidemia in both early- and late-onset infections. As a significant part of the human microbiota, *C. albicans* can cause BSI in cases of alterations in barrier functions, resulting in the translocation of flora [13,14].

*Candida parapsilosis*. In recent years, *C. parapsilosis* has been reported as the second most frequently isolated member of the *Candida* genus. The increasing incidence of *C. parapsilosis* early-onset BSI is related to the underlying co-morbidities requiring the presence of central venous catheters or the use of total parenteral nutrition [15]. Moreover, *C. parapsilosis* BSI has been associated with surgical procedures due to the extensive use of invasive techniques and insertion of indwelling devices [16].

*Candida tropicalis*. *C. tropicalis* is estimated to be the third or fourth most commonly isolated species among patients with candidemia. Compared to other species, *C. tropicalis* is more frequent in neutropenic patients with solid organ and hematological malignancies. Advanced age and chronic respiratory co-morbidity are also correlated to the high incidence of *C. tropicalis* BSI [17].

*Candida glabrata* (renamed to *Nakaseomyces glabrata*). Compared to other pathogens, *C. glabrata* is one of the main non-*albicans* species isolated in ICUs [18]. According to recent data, advanced age and prior azole exposure are associated with high prevalence *C. glabrata* BSIs among critically ill patients, resulting in worse outcomes. Since the incidence of fluconazole resistance has been increased, the management of patients with *C. glabrata* infection may be challenging [19,20,21].

*Candida krusei* (renamed to *Pichia kudriavzevii*). Rare but important, *C. krusei* may cause severe BSIs in immunocompromised patients due to its intrinsic resistance to fluconazole [22,23].

In the last decade, additional *Candida* species have become an important cause of fungal infection among patients in healthcare facilities. *Candida auris* is an emerging fungal pathogen associated with multi-drug resistance. Since its discovery in 2009, *C. auris* has been identified as a worldwide hazardous microorganism due to its potential to induce nosocomial outbreaks and invasive infections [24,25]. Although *C. auris* BSIs mainly occur in hospitalized patients rather than upon admission, it is worth mentioning in this review, as it poses a significant global health threat [26]. Unique characteristics of *C. auris* include its innate resistance to multiple antifungal agents, commonly to azoles or amphotericin B, and environmental persistence, as it can survive on surfaces and medical equipment for more than 3 weeks [27,28]. Moreover, there is a strong probability of misdiagnosed strains when standard laboratory systems are used. For accurate identification, molecular techniques or matrix-assisted laser desorption ionization time-of-flight mass spectrometry (MALDI-TOF-MS) technology is required [29]. The main clinical characteristics of the most common *Candida* species in the ICU environment are summarized in Table 1.

## 3. Risk Factors

In the environment of ICU, early-onset candidemia constantly raises concern due to the challenges in rapid diagnosis, as early infections are at higher risk of inappropriate treatment, resulting in an increased mortality rate [8]. Patients admitted to the ICU often have multiple risk factors that increase their vulnerability to *Candida* BSIs. Prompt identification of patients at risk is very important to ensure early, proper antifungal therapy [30]. A number of studies have identified several significant risk factors of early-onset candidemia in adult ICU patients, such as administration of broad-spectrum antibiotics for other infections, gastrointestinal perforation or recent gastrointestinal surgery, indwelling catheters, co-existing medical conditions, impaired immune function, major trauma and burns, total parenteral nutrition (TPN), the use of mechanical ventilation, and renal replacement therapy (Table 2) [31,32].

### 3.1. Administration of Broad-Spectrum Antibiotics

Patients admitted to the ICU frequently receive empiric broad-spectrum antibiotics for severe community-acquired infections. The bacterial microbiota of the gastrointestinal tract is often disrupted in these patients, where the imbalance between over- and under-represented microorganisms promotes *Candida* dissemination [33,34,35]. Several studies have investigated the association between prior antibiotic use and the development of candidemia [36,37]. A retrospective single-center case–control study has demonstrated the correlation between piperacillin/tazobactam and vancomycin use and subsequent *C. glabrata* and *C. krusei* candidemia [38]. Other categories of antibiotics found to be associated with fungal BSIs are cephalosporins, anti-anaerobic agents, glycopeptides, carbapenems, tigecycline, and aminoglycosides [39]. In general terms, broad-spectrum antibiotic use has been identified as a significant risk factor for invasive *Candida* infections in the ICU setting and specifically for candidemia [40,41].

### 3.2. Surgical and Non-Surgical Disorders of Gastrointestinal Tract

As mentioned above, *Candida* microorganisms commonly colonize the human gastrointestinal tract as a significant component of microbiota [42]. Patients admitted to the ICU after surgical interventions on the gastrointestinal tract are at higher risk for early-onset candidemia. Especially, conditions such as perforations, bowel anastomosis leaks, or peritonitis can lead to disbalance between fungal microorganisms and host, while loss of gastrointestinal tract continuity causes yeast translocation through the intestinal mucosal barrier and the entrance of the pathogens into the circulatory system [43,44,45]. Moreover, patients admitted to the ICU after surgical interventions on the gastrointestinal tract are frequently exposed to additional factors that favor the progression of *Candida* colonization, such as reduced bowel function, antibiotic use, and requirement for organ support [46].

**Table 2 diagnostics-15-01402-t002:** Common risk factors of early-onset candidemia in adult ICU.

Risk Factors	Characteristics
Broad-spectrum antibiotics [36,37,38,39,40,41]	Piperacillin/tazobactam, vancomycin, cephalosporins, anti-anaerobic agents, glycopeptides, carbapenems, and tigecycline, aminoglycosides
Disorders of the gastrointestinal tract [43,44,45,46,47]	Perforation, bowel anastomosis leaks, peritonitis, reduced bowel function, and acute pancreatitis
Indwelling catheters [48,49,50,51]	Central venous catheters, arterial catheters, and dialysis catheters
Co-morbidities [52,53,54,55,56]	Chronic cardiovascular disease, chronic respiratory disease, kidney dysfunction, and diabetes mellitus
Immunosuppression [57,58,59,60,61,62,63]	Septic shock, corticosteroid therapy, solid organ transplantation, hematopoietic stem cell transplantation, and diabetes mellitus
Major trauma and burns [64,65,66,67]	High injury severity score, increased number of blood transfusions, numerous surgical interventions, and immunosuppression
Total parenteral nutrition [30,68,69]	Use of central venous catheters, rapid proliferation in TPN solutions
Mechanical ventilation [47,70,71]	Requirement for more than 48 h increases the risk of invasive candidiasis
Renal replacement therapy [40,72]	Central venous catheters, recurrent manipulations

Likewise, patients with severe acute necrotizing pancreatitis may develop invasive candidiasis and candidemia due to progressive colonization of the gastrointestinal tract and *Candida* translocation into necrotic lesions and bloodstream [47].

### 3.3. Indwelling Catheters

Central venous catheters, arterial catheters, and dialysis catheters are inserted in most critically ill patients treated in the ICU. However, their extensive use has been associated with a high risk of *Candida* colonization and biofilm formation, resulting in early-onset bloodstream infection [48,49]. *Candida* species, mainly *C. albicans* and *C. parapsilosis,* are reported to form catheter-associated biofilms, which protect them from the host immune response and antifungal therapy [50]. In addition, the *Candida* genus is normally presented in human flora, whereas the installation of indwelling devices disrupts the physical barrier of the skin, allowing pathogens to access the bloodstream [51].

### 3.4. Co-Existing Medical Conditions

Early-onset candidemia in critically ill adults is often associated with various pre-existing clinical conditions that compromise immune system function and increase susceptibility to infectious pathogens. Notable co-morbidities linked to the high risk of early *Candida* BSIs in the ICU include chronic cardiovascular and respiratory diseases, kidney dysfunction, and diabetes mellitus [52,53].

Increased susceptibility to infections is a key clinical characteristic of diabetes mellitus. Inadequate control of glucose blood levels compromises neutrophil activity, resulting in altered immune response. Specifically, persistent hyperglycaemia reduces the release of pro-inflammatory cytokines and impairs neutrophil’s main anti-inflammatory mechanisms, such as recruitment, chemotaxis, phagocytosis, and intracellular reactive oxygen species (ROS) production [54]. Moreover, elevated glucose levels promote *Candida* biofilm formation on indwelling medical devices, providing a secure environment against antifungal therapeutic agents [55,56]. The combination of these factors increases the susceptibility of hyperglycaemic ICU patients to invasive *Candida* infections.

### 3.5. Immunosuppression

A number of immunosuppressive conditions have been connected with ICU-associated candidemia, such as septic shock, corticosteroid therapy, solid organ transplantation, hematopoietic stem cell transplantation, and diabetes mellitus.

Patients who develop septic shock and multiple organ dysfunction after their admission to the ICU are extremely vulnerable to fungal infections. Furthermore, neutropenia has been identified as an independent risk factor for mortality [57,58]. The administration of corticosteroids to patients with an ongoing requirement for vasopressor therapy is a common practice in the ICU, which further impairs immune response [59,60].

*Candida* species are the most common aetiological factors of fungal BSIs in solid organ transplant recipients. Patients who underwent liver, pancreatic, and small bowel transplantation are at high risk of candidemia, followed by heart and/or lung transplant recipients [61,62]. Likewise, delayed immune reconstitution after hematopoietic stem cell transplantation due to underlying disease and immunosuppressive treatment has been identified as a potential causative factor of early candidemia [63].

### 3.6. Major Trauma and Burns

Patients with major trauma and severe burns often require treatment at the ICU. Disruption of the epithelial barrier allows for deep tissue contamination with fungal pathogens, while extensive use of broad-spectrum antibiotics, high injury severity score, increased number of blood transfusions, numerous surgical interventions, and immunosuppression stand for important predictors of early-onset candidemia [64,65,66,67]. Candidemia occurring within 96 h after admission constitutes an uncommon but not rare complication of multiple trauma.

### 3.7. Total Parenteral Nutrition

TPN has been well documented as a significant risk factor for candidemia [68]. Recent data suggested that *Candida* species can proliferate rapidly in almost all TPN solutions, regardless of their acidity or lipid and sodium bisulfate content [69]. Additionally, the use of central venous lines for TPN delivery promotes *Candida* colonization and biofilm formation, as highlighted above. A prospective multicenter matched case–control study emphasized that TPN administration increased the risk of developing candidemia in both ICU and non-ICU populations [30].

### 3.8. Mechanical Ventilation

*Candida* colonization of the respiratory tract is prevalent (approximately 27%) among patients receiving invasive mechanical ventilation for more than 48 h. *C. albicans* is the most common isolated pathogen, followed by *C. glabrata* and *C. tropicalis* [70]. While the identification of *Candida* microorganisms in respiratory specimens does not confirm infection, it appears as a significant risk factor for invasive candidiasis and candidemia in critically ill patients [47,71].

### 3.9. Renal Replacement Therapy

Patients in the ICU frequently receive continuous renal replacement therapy or haemodialysis as part of the management of the underlying disease. Central venous catheter placement and recurrent manipulations on it are associated with an increased risk of early *Candida* BSIs in critically ill patients [40,72].

## 4. Clinical Manifestations

Early-onset candidemia in adult ICU patients generally presents with non-specific clinical manifestations, which are often overlapped by symptoms of the underlying disease. Thus, it is essential for clinicians to be highly aware, when evaluating patients with aforementioned risk factors [73].

Depending on the patient’s condition, the clinical presentation of candidemia can range from the absence of specific symptoms to severe septic shock [74,75]. Common signs include persistent fever and rigor, and unresponsive to broad-spectrum antibiotic therapy. In critically ill patients early-onset candidemia may initially present with clinical signs of sepsis and septic shock, including hypotension, tachycardia, and tachypnoea. The dissemination of the *Candida* pathogen through the bloodstream may lead to the dysfunction of multiple organs [76]. In general, septic shock due to candidemia presents significant treatment challenges, demonstrating a delayed response to basic therapeutic approaches to sepsis, such as vasopressors and antimicrobials [77].

Failure in prompt diagnosis and delayed initiation of antifungal therapy may lead to hematogenous spread of the *Candida* pathogen and secondary infections in other organs, resulting in organ-specific symptoms. All patients with candidemia are susceptible to endogenous endophthalmitis; thus, a screening ophthalmological examination is highly recommended in these cases [78,79]. The development of a new heart murmur, changes in the quality of a pre-existing murmur, or end-organ symptoms from embolic phenomena may suggest infectious *Candida* endocarditis [80,81]. Immunocompromised patients with a neutrophil count below 500 μL, fever, and abnormal liver biochemical tests should be evaluated further for hepatosplenic candidiasis [82]. Patients with renal candidiasis due to candidemia may present with high fever, haemodynamic instability, and variable renal insufficiency [83]. Finally, meningitis/ventriculitis or cerebral abscess due to *Candida* species are usually associated with the presence of indwelling neurosurgical devices. Patients with changes in mental status, new-onset seizures, and neurological deficits should be examined for this uncommon, but serious complication of *Candida* BSIs [84,85]. Given the heterogeneous clinical presentation, maintaining a high index of suspicion is crucial for the detection and management of early-onset candidemia in ICU patients.

## 5. Diagnosis

Early-onset candidemia in adult ICU patients presents significant diagnostic challenges due to non-specific clinical signs, symptoms, and the limitations of conventional diagnostic techniques [86]. The gold standard for diagnosing candidemia remains the isolation of the *Candida* pathogen from blood cultures, where the positive result should always be considered as an actual infection and not a contaminant [87]. Other diagnostic approaches include non-culture-based methods, such as serum biomarkers and molecular biology. The main laboratory tests used for the diagnosis of candidemia are summarized in Table 3. Combining clinical and laboratory assessment with predictive scoring systems may enhance diagnostic accuracy. A possible diagnostic algorithm for critically ill patients at risk for early-onset candidemia is presented in Figure 1.

### 5.1. Blood Cultures

Blood cultures are traditionally considered the key to a definitive diagnosis of candidemia. However, there are notable limitations regarding sensitivity and the time to identification of the pathogen. Depending on the results from various studies, the sensitivity of blood cultures to detect *Candida* ranges from 51% to 71%, with lower rates in neutropenic patients [88,89,90]. Type of candidemia (primary BSI or candidemia secondary to deep-seated infection), type of *Candida* species *(C. glabrata* has a longer incubation period), and previous exposure to antifungal agents may affect the sensitivity of blood culture, leading to false negative results. Increasing the volume of the complete set of blood cultures to 60 mL and blood collection every 24 h may enhance the sensitivity of the method [90].

The major limitation of blood culture is the prolonged time to detect and identify *Candida* pathogens. Depending on the *Candida* species, detecting yeast growth may require 24–72 h, followed by an additional 24–48 h for identification. Current data showed that *C. glabrata* has significantly longer detection and identification time compared to other species, resulting in a longer time to the administration of appropriate antifungal therapy [91,92].

In the case of a positive blood culture, various analytic techniques have been used to reduce the time required to identify *Candida* subculture. Matrix-assisted laser desorption ionization time-of-flight mass spectrometry (MALDI-TOF-MS) technology offers a considerably shorter time for *Candida* identification in the blood culture broth, with a reported sensitivity of 56–73%. Moreover, MALDI-TOF-MS can be used for analysis of antifungal susceptibility [93,94,95]. Peptide nucleic acid-fluorescence in situ hybridization (PNA-FISH) may provide *Candida* identification and an antifungal susceptibility panel within several hours after blood culture positivity [96,97,98]. It is essential for the management of patients with candidemia to perform antifungal susceptibility tests after pathogen identification. Increased resistance to azoles and echinocandins is a serious issue worldwide, while inappropriate antifungal therapy may result in an increased risk of death for patients [99,100,101].

### 5.2. Molecular Techniques

Blood culture-independent molecular assays provide an alternative approach for rapid detection of *Candida* species. These advanced methodologies offer improved sensitivity, specificity, and speed. They do not require positive blood cultures and can be performed directly on whole blood, serum, or plasma samples [102].

Polymerase chain reaction (PCR)-based test is a cornerstone molecular technique that enables the detection of *Candida* DNA directly from clinical specimens with high sensitivity (90–95%) and specificity (90–92%) [103,104]. Multiple *Candida* PCR panels, accessible for analysis, allow for the rapid identification of the pathogen, even in cases where blood cultures report negative results [105,106]. More complex real-time PCR (qPCR) allows for the monitoring of DNA amplification in real time, based on both detection and measurement of DNA load [107]. A nucleic acid sequence-based amplification (NASBA) assay is an isothermal amplification method that targets RNA. This sensitive technique provides rapid detection and identification of medically important *Candida* species [108]. Another rapid molecular assay, based on loop-mediated isothermal amplification (LAMP), uses a robust DNA polymerase that catalyzes high-speed amplification without DNA purification. This simple and rapid procedure, which is currently available for research use only, may optimize the management of patients with early-onset candidemia [109]. The novel genome-editing technique CRISPR (short for “clustered regularly interspaced short palindromic repeats”) has been adapted for the detection of *Candida* species, providing easy operation, low setup cost, and high sensitivity and specificity. When combined with conventional isothermal amplification methods, the CRISPR-based diagnostic platform offers rapid and accurate identification of *Candida* pathogens. Despite their advantages, CRISPR-based diagnostic platforms are still in limited use due to challenges like the need for sample processing, the cold-chain requirement during storage and transport, the need for specialized equipment, and deficiency in low-resource settings [110,111].

An innovative approach for diagnosis of *Candida* infection has been introduced in the last few years, using a combination of miniaturized magnetic resonance technology with PCR-based assay. A T2Candida panel does not require blood culture or nucleic acid extraction to provide rapid, sensitive, and specific detection of *Candida* in whole blood samples. Specifically, the platform allows for the identification of five of the most medically important *Candida* species (*albicans*, *tropicalis*, *parapsilosis*, *krusei*, and *glabrata*) within 3–5 h and without the need for prior preparation of the sample [112,113,114].

While all the molecular techniques described provide high accuracy, sensitivity, and specificity to detect *Candida* pathogens, none of them can distinguish between contamination and ongoing infection. Thus, diagnostic stewardship is strongly recommended in patients with suspected *Candida* BSI.

### 5.3. Serum Biomarkers

Serum biomarkers play an essential role in the prompt diagnosis of early-onset candidemia in the ICU, especially when results from blood cultures are delayed. Similarly to molecular techniques, available biomarkers are not validated to discriminate colonization from ongoing infection, despite the extensive progress in research [115]. Key biomarkers include 1,3-β-D-glucan (BDG), mannan antigen, anti-mannan antibody, and *Candida* species germ tube antibody (CAGTA).

The BDG assay detects 1,3-β-D-glucan, a cell wall polysaccharide of most fungi, including *Candida* species. Current data suggest manifold diagnostic accuracy of the assay, depending on co-morbidities, risk factors, type of infections, and the cut-off value used to determine positivity [116,117]. A high diagnostic sensitivity of 70–80% and a specificity of approximately 55–80% have been reported, with the best results in high-risk patient groups [104,116]. The BDG assay is noted for its excellent negative predictive value, allowing for the exclusion of invasive fungal infections and the management of antifungal therapy (avoidance or discontinuation). Moreover, a combination of BDG tests with predictive scoring systems or other fungal serum biomarkers optimizes its application in clinical practice [118,119,120]. The greatest limitation of the BDG assay is its poor species specificity. Furthermore, patients with albumin, intravenous immunoglobulin, certain antibiotics, bacterial infections, or renal replacement therapy with cellulose membranes may have false positive results [121,122].

Mannans are a class of *Candida* cell wall polysaccharides used for the detection of BSI. During infection, *Candida* releases mannan into the bloodstream, introducing a diagnostic technique based on the detection of mannan antigen and the host’s anti-mannan antibodies [123]. When used together, the detection of both mannan antigens and antibodies increases sensitivity and specificity [51,124]. However, a preplanned ancillary analysis of the EMPIRICUS randomized clinical trial demonstrated poor predictive accuracy of mannan antigen and anti-mannan antibodies detection in ICU patients [125]. A combination of these tests with other serum biomarkers may exhibit sensitivity and improve the diagnostic efficacy of early-onset candidemia [126].

**Table 3 diagnostics-15-01402-t003:** Main features of different methods used for diagnosis of candidemia.

Laboratory test	Characteristics
Blood cultures [88,89,90,91,92,93,94,95,96,97,98,99,100,101]	Definitive diagnosis Susceptibility testing Long turnaround time (reduced with MALDI-TOF-MS and PNA-FISH technology)
PCR-based tests [103,104,105,106,107,108,109,110,111]	Rapid turnaround time High sensitivity and specificity Multiple PCR panels High cost
T2Candida panel [112,113,114]	Combination of methods High sensitivity and specificity Rapid turnaround time No need for sample preparation High cost of purchasing equipment for the laboratory Not available in all laboratories
BDG assay [104,116,117,118,119,120,121,122]	High diagnostic sensitivity Rapid turnaround time High negative predictive value Poor species specificity
CAGTA [124,127,128,129,130]	Rapid turnaround time Better detection of candidemia and deep-seated candidiasis
Mannan antigen and anti-mannan antibody [51,123,124,125,126]	Rapid turnaround time Increased sensitivity and specificity when used together Suboptimal predictive accuracy

The detection of CAGTA is a technique that relies on laboratory identification of antibodies produced against specific superficial antigens of the germ tubes of *C. albicans* by indirect immunofluorescence [127]. CAGTA detection is particularly useful in diagnosing *Candida* BSIs and deep-seated candidiasis in critically ill or immunocompromised patients [128,129]. According to different reports, the sensitivity and specificity of the method range from 42% to 96% and 54% to 100%, respectively, depending on the type of infection, with better values noted in candidemia. When combined with BDG tests or mannan antigens/antibodies, CAGTA detection improves diagnostic accuracy [124,130].

### 5.4. Predictive Scoring Systems

A late diagnosis of early-onset candidemia and a delayed initiation of antifungal therapy further increase mortality in the ICU [131]. Most critically ill patients exhibit non-specific clinical presentation of *Candida* BSI. Moreover, current pathogen detection techniques have various limitations regarding the turnaround time and diagnostic accuracy. Therefore, the prompt recognition of early-onset candidemia is a major issue in the ICU population [132]. Predictive scoring systems help identify ICU patients at high risk for candidemia, enabling early and efficient antifungal treatment. (Table 4) These particular models integrate clinical risk factors, laboratory markers, and patients’ histories to distinguish between patients at low and high risk of infection.

“Candida Score” is a simple bedside scoring system that may assist clinicians in distinguishing colonization from fungal infection in non-neutropenic critically ill patients. Scoring criteria include multifocal *Candida* colonization, surgery on ICU admission, total parenteral nutrition, and severe sepsis. With a cut-off value of 2.5, “Candida score” has reported a sensitivity of 81% and specificity of 74%, which may help the intensivists to identify patients who will benefit from early antifungal administration [133].

The Candida colonization index was described in 1994 as a predictive model of subsequent *Candida* infections in critically ill surgical patients [134]. The Candida colonization index is defined as the ratio of the number of non-blood body cites colonized by *Candida* species to the total number of cultured cites, with a cut-off value of 0.5. The sensitivity and specificity of the method were determined to be 100% and 69%, respectively. A significant negative predictive value may assist in the surveillance of colonization dynamics in ICU patients [47,135].

The Ostrosky-Zeichner score is a broader risk assessment tool, which helps identify patients at risk for invasive candidiasis in the ICU using logistic regression models. According to the authors, the main risk factors are a need for mechanical ventilation, central venous catheters, use of broad-spectrum antibiotics, parenteral nutrition, dialysis, major surgery, pancreatitis, steroid, and immunosuppressive therapy [136,137,138]. The combination of certain criteria was connected to an increased risk of invasive candidiasis in the intensive care setting.

In general, predictive scoring systems play a significant role in identifying high-risk ICU patients for early-onset candidemia and guiding prompt antifungal therapy, especially when used in combination with other diagnostic techniques. However, these methods have several notable limitations, including limited sensitivity, delayed recognition of BSI, and limited predictive power. Furthermore, the Candida colonization index requires time-consuming extensive microbiological surveillance, which makes early recognition of *Candida* infection actually challenging. In addition, many critically ill patients meet the criteria used in the calculation of the Ostrosky-Zeichner score, without any evidence of *Candida* colonization or infection, making it less reliable in a general ICU environment.

## 6. Management

Managing early-onset candidemia in adult ICU patients requires a multidisciplinary approach involving early diagnosis, appropriate empirical and targeted antifungal therapy, source control, and general supportive care, as most patients with *Candida* BSIs present with septic shock and hemodynamic instability [139,140]. Therapeutic strategies for candidemia include prophylaxis, pre-emptive therapy, empirical, and targeted antifungal treatment [141] (Table 5).

### 6.1. Antifungal Therapy

Antifungal prophylaxis is the treatment of patients at high risk for the development of candidemia, without any signs or symptoms of fungal disease. The purpose of prophylactic therapy among critically ill patients is to reduce morbidity and mortality by decreasing fungal overload [142,143]. Several studies have assessed the use of prophylactic treatment in critically ill patients. A meta-analysis of randomized controlled trials demonstrated that the administration of azoles (fluconazole, ketoconazole) reduced the prevalence of invasive fungal infections, despite the heterogeneity in clinical and methodological characteristics [144]. However, the impact of prophylactic treatment on mortality is still debatable. Current guidelines recommend against the routine and universal use of antifungal prophylaxis in the ICU population [145]. Nonetheless, there are certain groups of patients who may benefit from prophylactic therapy, such as those in surgical ICUs with recent abdominal surgery and recurrent gastrointestinal perforations or anastomotic leakages, neutropenic patients, solid organ transplant recipients and patients undergoing allogeneic haematopoietic stem cell transplant [131,146]. Fluconazole at a dose of 400 mg IV/PO daily is preferred in patients without prior azole exposure and in environments where *C. albicans* predominates. In ICUs with high rates of fluconazole-resistant *Candida* species (*C. glabrata* or *krusei*), echinocandins are recommended. For prophylactic use, caspofungin may be administered at a dose of 50 mg IV daily or micafungin at a dose of 100 mg IV daily [147,148]. Resistance to azoles, resulting in the emergence of non-*albicans* species, constitutes an important side effect of uncontrolled use of antifungal prophylaxis [149]. Moreover, some ICU patients may never develop candidemia, making routine prophylaxis unnecessary. While the high cost of echinocandins further attenuates their broad use as prophylactic treatment [11,150].

Pre-emptive antifungal treatment targets a group of critically ill patients at high risk for candidemia who have positive laboratory or microbiological evidence of *Candida*, but no clinical signs of infection [151]. In some cases, it serves as a midpoint between empirical and targeted treatment. This approach has been supported by a number of clinical studies; however, the criteria for the initiation of pre-emptive therapy remain unclear [152]. Patients with a high-risk profile on admission or high *Candida* colonization burden and positive biomarkers (BDG, mannan antigens/antibodies, or PCR) may benefit from pre-emptive antifungal treatment. Agents used for pre-emptive therapy are the same as in prophylaxis. However, there are no universal guidelines for the administration of pre-emptive therapy in critically ill patients, as current data present no significant difference in mortality [145,153]. Additionally, there are no certain acceptable thresholds for initiating therapy. The protocols of every ICU vary depending on the availability of molecular assays and biomarker detection tests [154]. As mentioned above, BDG may provide false positive results in the presence of certain clinical conditions, such as several antibiotics, bacterial infections, or hemodialysis; therefore, some patients may receive unnecessary antifungal treatment.

**Table 5 diagnostics-15-01402-t005:** Therapeutic strategies for early-onset candidemia in the ICU.

Strategy	Antifungal Agents	Comments	References
Prophylaxis	- Fluconazole: 400 mg IV/PO daily - Caspofungin: 50 mg IV daily - Micafungin: 100 mg IV daily	- High-risk patients - Azoles are preferred in patients without prior azole exposure	Cornely, O.A. et al. [131] Cornely, O.A. et al. [142] Echeverria, P. et al. [143] Martin-Loeches, I. et al. [145] Einav, S. et al. [146] Cortegiani, A. et al. [147] Chen, S.C.A. et al. [148]
Pre-emptive therapy	- Fluconazole: 400 mg IV/PO daily - Caspofungin: 50 mg IV daily - Micafungin: 100 mg IV daily	- Patients with a high-risk profile on admission or high *Candida* colonization burden and positive serum biomarkers	Martin-Loeches, I. et al. [145] Sprute, R. et al. [151] Pham, H.T. et al. [153]
Empirical therapy	- Caspofungin: loading dose 70 mg IV, then 50 mg IV daily - Micafungin: 100 mg IV daily - Anidulafungin: loading dose 200 mg IV, then 100 mg IV daily - Fluconazole: 800 mg IV loading dose, then 400 mg IV daily - Liposomal Amphotericin B 3–5 mg/kg IV daily	- High-risk patients with clinical signs of infection - Echinocandins as first-line therapy - Azoles are an acceptable alternative in stable patients without prior azole exposure - Liposomal Amphotericin B is recommended if there is intolerance or resistance to other antifungal agents	Pappas, P.G. et al. [73] Martin-Loeches, I. et al. [145] León, C. et al. [155] Klastersky, J. [156] Tang, B.H.E. et al. [157] Kanj, S.S. et al. [158]
Targeted therapy	- Caspofungin: loading dose 70 mg IV, then 50 mg IV daily - Micafungin: 100 mg IV daily - Anidulafungin: loading dose 200 mg IV, then 100 mg IV daily - Fluconazole: 800 mg IV loading dose, then 400 mg IV daily - Liposomal Amphotericin B 3–5 mg/kg IV daily	- Patients with clinical signs and microbiologically approved infection - Species specific and susceptibility-guided treatment - Echinocandins as first-line therapy - Azoles as step-down therapy, or first-line therapy in stable patients without suspected azole resistance - Liposomal Amphotericin B is recommended if there is intolerance or resistance to other antifungal agents	Pappas, P.G. et al. [73] Martin-Loeches, I. et al. [145] Boutin, C.A. et al. [159] Chatelon, J. et al. [160] Garnacho-Montero, J. et al. [161] Yang, Q. et al. [162]

Empirical treatment refers to the administration of antifungal agents to patients with suspected candidemia, before microbiologic confirmation, based on clinical signs of infection and risk factors [155]. Empirical therapy is widely used in the environment of the ICU, due to non-specific clinical signs of infection and delayed blood culture results [156]. Current guidelines recommend the use of empirical antifungal therapy only in patients with high-risk factors for *Candida* BSI, who present with septic shock and multiple organ failure and have more than one site positive for *Candida* colonization [73,145]. The preferred therapeutic choice for non-neutropenic critically ill patients is echinocandins (caspofungin: loading dose of 70 mg, then 50 mg daily; micafungin: 100 mg daily; anidulafungin: loading dose of 200 mg, then 100 mg daily). However, in stable patients without previous azole exposure and the absence of azole-resistant *Candida* species in colonized sites, fluconazole might be a reasonable option at a loading dose of 800 mg, followed by 400 mg daily. Finally, in rare cases, when the suspected pathogen is considered resistant to both the abovementioned groups of antifungal agents, liposomal Amphotericin B may be initiated at a dose of 3–5 mg/kg daily [73,157]. In order to limit unnecessary use, de-escalation or termination of empirical antifungal therapy is recommended in 4–5 days when there is no clinical response or no evidence of microbiological, molecular, or laboratory confirmation of candidemia. Published data demonstrate contradictory results regarding the improvement of hospital mortality rate. However, some patients may benefit from early empirical antifungal therapy [158]. As soon as the *Candida* pathogen is confirmed in blood cultures, empirical treatment must be adjusted and continued as targeted therapy.

Targeted therapy ensures appropriate, species-specific, and susceptibility-guided treatment [159]. Regarding non-neutropenic critically ill patients, Infectious Diseases Society of America (IDSA) guidelines and European Society of Intensive Care Medicine (ESICM)/Critically Ill Patients Study Group of the European Society of Clinical Microbiology and Infectious Diseases (ESCMID) guidelines recommend administration of echinocandins as first-line antifungal therapy [160]. Fluconazole can primarily be used as a step-down therapy in less severe patients (absence of septic shock or multiple organ failure) when the isolated pathogen is susceptible to azoles. Liposomal Amphotericin B in a dose of 3–5 mg/kg daily must be reserved for resistant infections or if intolerance to first-line antifungal agents occurs [73,161]. Additionally, a combination of antifungals has been attempted in patients in ICUs (caspofungin with voriconazole) with no significant differences in outcomes [162]. Follow-up blood cultures must be performed every day or every 48 h prior to negative results. For uncomplicated candidemia, treatment should continue for 14 days after the documented first negative blood culture [145]. Extended therapy duration is necessary in cases of persistent candidemia or when the source of infection cannot be eliminated.

The extensive use of antifungal agents in ICU patients is associated with high rates of adverse reactions and leads to the development of antifungal resistance [163]. The antifungal stewardship (AFS) program is essential to optimize antifungal use, improve patient outcomes, and attenuate the development of resistance [164]. Effective AFS programs utilize non-culture-based diagnostic methods and molecular techniques to guide early initiation or discontinuation of antifungal therapy. Routine susceptibility testing is recommended to inform treatment decisions and support de-escalation strategies [165,166]. Other essential elements of a successful AFS program involve multidisciplinary collaboration, developing institution-specific guidelines based on local epidemiology and resource availability, and providing continuous education. Finally, regular monitoring of antifungal prescriptions helps identify controversial areas for improvement and supports stewardship efforts [167].

### 6.2. Source Control

Effective management of early-onset candidemia requires prompt source control, which involves identifying and eliminating the origin of the infection. Current data demonstrate an improved clinical outcome among patients who received appropriate antifungal treatment in combination with early source control [86,168,169]. Key points of source control in candidemia include removal of central venous catheters, management of other prosthetic devices, and drainage of infected sites [145]. Current guidelines strongly recommend the early removal of central lines, especially in cases when they are suspected to be the origin of infection and can be removed safely [73,170]. In cases of *Candida* BSI secondary to deep-seated infection, source control may require percutaneous drainage and debridement of infected tissues [145,171]. Finally, in some patients, other indwelling devices beyond central venous catheters may be implicated in the infection; thus, clinicians may consider removing or replacing these implants in order to achieve source control [172,173,174].

### 6.3. General Supportive Management

Effective hemodynamic and supportive care is crucial in critically ill patients with early-onset candidemia, particularly in those with signs of septic shock. Adequate fluid resuscitation and early administration of vasopressors (norepinephrine as first-line vasopressor) are recommended, targeting mean arterial pressure of over 65 mm Hg [59]. Overall, early effective antifungal treatment with adequate source control represents the best approach for improving the survival of patients with septic shock due to candidemia [175].

## 7. Prognosis and Prevention

The prognosis of early-onset candidemia in adult ICU patients relies on several important factors, including the timing of diagnosis and initiation of antifungal therapy, *Candida* species, severity of the patient’s condition, and adequacy of source control. Postponement of diagnosis and treatment for over 48 h significantly increases mortality in these patients, along with the administration of inappropriate antifungal therapy [8,176]. The severity of clinical manifestations, presence of septic shock, high Acute Physiology and Chronic Health Evaluation II (APACHE II) score, and Sequential Organ Failure Assessment (SOFA) Score are also correlated with increased mortality [177,178,179,180,181,182]. The *Candida* pathogen also affects prognosis, as patients with isolated *C. albicans* have better outcomes than those with *C. glabrata* or *parapsilosis* [183,184]. Inadequate source control has been identified as a significant risk factor for mortality among critically ill patients with candidemia. Studies have demonstrated that the maintenance of infected catheters and devices, or incomplete abscess drainage, worsens prognosis, highlighting the importance of early and appropriate intervention [169,177,185,186].

*Candida* BSIs remain a major public health problem, especially in critically ill patients. The global burden of invasive Candida infections in the ICU environment is increasing. Among critically ill patients, two-thirds present with candidemia. Worldwide, candidemia ranges from 5.5 to 7 episodes per 1000 ICU admissions, according to recent reports. Over the past decade, a shift from more susceptible *Candida* species (*C. albicans*) to the less susceptible or resistant non-*albicans Candida* species has been observed. Especially *C. auris* shares unique characteristics, including multidrug resistance and persistence in patient flora and healthcare environment [14,76,187]. Despite the improved prognosis and advanced therapeutic approaches, candidemia in the ICU setting is associated with high mortality rates, typically ranging from 30% to 60% [188,189,190].

Given the high morbidity and mortality rates, prevention of *Candida* infections is crucial and involves both general preventive ICU practices and approved antifungal strategies. Main infection control measures include the use of strict aseptic techniques during central catheter management and early removal of unnecessary lines [191]. Since multiple-site colonization with *Candida* pathogens is commonly recognized as a risk factor for candidemia, regular surveillance is necessary in critically ill patients to evaluate those at high risk and to prevent transmission [135]. Furthermore, limiting the use of broad-spectrum antibiotics can reduce the risk of *Candida* overgrowth [192]. Following these infection control strategies may significantly reduce the incidence of early-onset candidemia and improve patient outcomes in the ICU environment.

## 8. Conclusions

Early-onset candidemia commonly occurs within the first 7 days after ICU admission and remains a serious and life-threatening infection among critically ill patients. Clinical characteristics and risk factors may differ between early- and late-onset infections, resulting in misdiagnosis and inappropriate treatment. Critically ill patients with multiple risk factors, such as the administration of broad-spectrum antibiotics, central venous catheters and indwelling devices, recent gastrointestinal surgery, immunosuppression, mechanical ventilation, major trauma and burns, total parenteral nutrition, and renal replacement therapy are at high risk for the development of early-onset candidemia. Although the mortality rate in early-onset infections is lower compared to late-onset infections, diagnostic delays and challenges in differentiating it from bacterial infections are significantly associated with patients’ survival. A multidisciplinary approach that integrates a high index of suspicion, early recognition, prompt initiation of appropriate antifungal therapy, and effective source control can significantly improve prognosis and outcomes. The accurate application of prevention methods and the implementation of antifungal stewardship practices play a critical role in reducing the incidence and impact of early-onset candidemia.

## Figures and Tables

**Figure 1 diagnostics-15-01402-f001:**
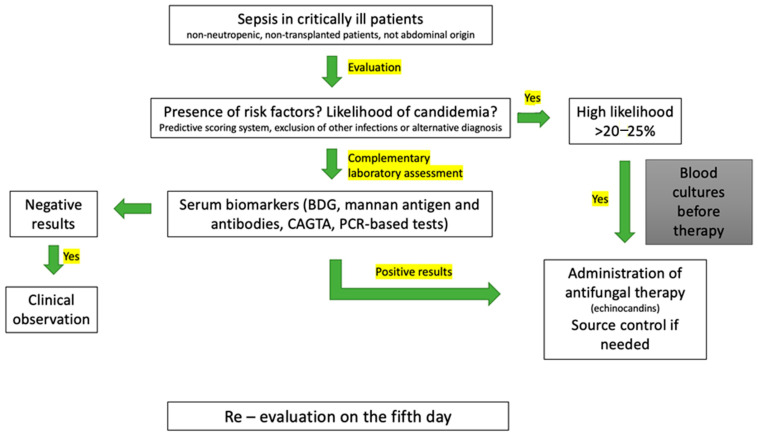
The proposed diagnostic algorithm in critically ill patients at risk for early-onset candidemia.

**Table 1 diagnostics-15-01402-t001:** Clinical characteristics of the most common *Candida* species in ICU.

Species	Age Predisposition	Prevalence in ICU	Virulence Factors	Common Risk Factors	Antifungal Resistance	Notable Clinical Associations
*C.albicans* [13,14]	None	Common	Biofilm formation, tissue invasion	Broad-spectrum antibiotics, central venous catheters	Generally susceptible	High dissemination risk, deep organ invasion
*C. parapsilosis* [15,16]	Younger	Increasing	Biofilm formation	Central venous catheters, TPN, surgical procedures	Reduced echinocandin susceptibility, Azole-sensitive	TPN, central lines, indwelling devices
*C. tropicalis* [17]	None	Common in neutropenic and cancer patients	Hyphal formation, deep tissue invasion	Neutropenia, malignancies, advanced age, chronic respiratory co-morbidity	Variable azole resistance, echinocandin-sensitive	Aggressive course, organ dissemination
*C. glabrata* [18,19,20,21]	Advanced	Increasing	Adherence	Advances in age, prior azole exposure, diabetes, immunosuppression	Often resistant to fluconazole, variable to echinocandins	High mortality requires susceptibility-guided therapy
*C. krusei* [22,23]	None	Less common	Moderate	Immunosuppression	Fluconazole-resistant, echinocandin-sensitive	Requires susceptibility-guided therapy
*C. auris* [24,25,26,27,28,29]	None	Increasing	Biofilm formation, hyphal formation, adherence	Antibiotic use, medical devices, immunosuppression, frequent hospitalization in healthcare facilities	Often resistant to fluconazole, variable to Amphotericin B	Nosocomial outbreaks, invasive infections, requires susceptibility-guided therapy

ICU: intensive care unit, TPN: total parenteral nutrition.

**Table 4 diagnostics-15-01402-t004:** Predictive scoring systems used in the ICU setting.

Scoring System	Characteristics	References
Candida Score - Surgery on ICU admission (1 point) - Total parenteral nutrition (1 point) - Septic shock (2 points) - Multiple site *Candida* colonization (1 point)	>2.5 points: predictor of invasive candidiasis Sensitivity 81%, specificity of 74%	León, C. et al. [133]
Candida Colonization Index - Ratio of the numbers of non-blood body cites colonized by *Candida* species to the total number of cultured cites	A score> 0.5 is considered positive Sensitivity 100%, specificity 69%	Eggimann, P. et al. [47] Pittet, D. et al. [134] Caggiano, G. et al. [135]
Ostrosky-Zeichner score - Broad-spectrum antibiotics (days 1–3) OR - Central venous catheter (days 1–3) AND at least 2 of the following factors - Mechanical ventilation- Total parenteral nutrition (days 1–3) - Dialysis (days 1–3) - Major surgery (days −7–0) - Pancreatitis (days −7–0) - Steroid and immunosuppressive therapy (days −7–0)	High negative predictive value Sensitivity 70%, specificity 60%	Ostrosky-Zeichner, L. et al. [136] Hermsen, E.D. et al. [137] Ostrosky-Zeichner, L. et al. [138]

## Data Availability

All data analyzed during this study are included in this published article.

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
