# Peer review of "Early-Onset Candidemia in Adult Intensive Care Units"

_diagnostics, 2025, doi:10.3390/diagnostics15111402_

Round 1
Reviewer 1 Report
Comments and Suggestions for Authors
Mouratidou and colleagues submit a review of early-onset candidemia in adult ICUs.
Comments:
- Latin genus and species names should be italicized, unless this particular journal has a unique style.
- Rephrase "Last decade, additional Candida species has..." to "In the last decade, additional Candida species have...".
- Table 1 should have a footnote containing the abbreviations used within the table, unless the journal style of having the abbreviation table at the end of the manuscript is acceptable to the journal.
- Rephrase "Patients developed..." to "Patients who developed...".
- Patients who have had small bowel transplantation are at risk of candidemia.
- Rephrase "prevalent (approximately 26.6%)" to "prevalent (approximately 27%)".
- In Table 3, one of the limitations of the T2Candida panel is the up-front high cost of purchasing the equipment for the laboratory.
- In Table 3, what is "Betted detection"?
- This reviewer feels that CRISPR-based diagnostic platforms are in extremely limited use; please provide some information regarding whether they are used at all in the real world.
- Add references to Table 4 for the scoring systems and their sensitivity and specificity values, perhaps as a third column. The columns should be labelled.
- Add a column to Table 5 with references.
- Rephrase "midpoint between empirical ant targeted treatment" to
midpoint between empirical and targeted treatment". - "Anyhow" is not a good word to use in publications. Please edit it out of any sentences where it is used.
- Check the references for missing page numbers. For many of the older publications, there is only a single page number, when perhaps there should be a range.
Author Response
Comments:
- Latin genus and species names should be italicized, unless this particular journal has a unique style.
- Rephrase "Last decade, additional Candida species has..." to "In the last decade, additional Candida species have...".
- Table 1 should have a footnote containing the abbreviations used within the table, unless the journal style of having the abbreviation table at the end of the manuscript is acceptable to the journal.
- Rephrase "Patients developed..." to "Patients who developed...".
- Patients who have had small bowel transplantation are at risk of candidemia.
- Rephrase "prevalent (approximately 26.6%)" to "prevalent (approximately 27%)".
- In Table 3, one of the limitations of the T2Candida panel is the up-front high cost of purchasing the equipment for the laboratory.
- In Table 3, what is "Betted detection"?
- This reviewer feels that CRISPR-based diagnostic platforms are in extremely limited use; please provide some information regarding whether they are used at all in the real world.
- Add references to Table 4 for the scoring systems and their sensitivity and specificity values, perhaps as a third column. The columns should be labelled.
- Add a column to Table 5 with references.
- Rephrase "midpoint between empirical ant targeted treatment" to
midpoint between empirical and targeted treatment". - "Anyhow" is not a good word to use in publications. Please edit it out of any sentences where it is used.
- Check the references for missing page numbers. For many of the older publications, there is only a single page number, when perhaps there should be a range
Response: Please see the attachment

Reviewer 2 Report
Comments and Suggestions for Authors
This review extensively described the candidemia in intensive care unit and emphasized on incorporating antifungal stewardship program in clinical practice. Review is well written and presented with updated diagnostics, biomarkers and management of early-onset candidemia in adult ICU patients. To improve further please consider below stated suggestions
Review is missing global burden/statistics of candidemia in intensive care unit
Consider to cite the references for the information in the tables
Line no. 247, correct the neutrophil count below 500 μL.
Author Response
Comments:
This review extensively described the candidemia in intensive care unit and emphasized on incorporating antifungal stewardship program in clinical practice. Review is well written and presented with updated diagnostics, biomarkers and management of early-onset candidemia in adult ICU patients. To improve further please consider below stated suggestions
Review is missing global burden/statistics of candidemia in intensive care unit
Consider to cite the references for the information in the tables
Line no. 247, correct the neutrophil count below 500 μL.
Response: Please see the attachment.
